



# Global soil NO emissions for Atmospheric Chemical Transport Modelling: CAMS-GLOB-SOIL v2.2

David Simpson[1,2] and Sabine Darras[3]

[1]EMEP MSC-W, Norwegian Meteorological Institute, Oslo, Norway
[2]Dept. Space, Earth & Environment, Chalmers Univ. Technology, Gothenburg, Sweden
[3]Observatoire Midi-Pyrénées, Toulouse, France

**Correspondence:** D. Simpson
(david.simpson@met.no)

**Abstract.**

We present a dataset of global soil NO emissions comprising gridded monthly data and the corresponding 3-hourly weight factors, suitable for atmospheric chemistry modelling. Data are provided globally at $0.5°\times0.5°$ degrees horizontal resolution, and with monthly time resolution over the period 2000-2018. Emissions are provided as total values and also with separate data
for soil NO emissions from background biome values, and those induced by fertilizers/manure, pulsing effects, and atmospheric deposition, so that users can include, exclude or modify each component if wanted.

This paper presents the emission algorithms and their data-sources, some comments on the availability of soil NO emissions in other inventories (and how to avoid double-counting), and finally some preliminary modelling results and comparison with observed data.

This dataset was constructed as part of the Copernicus Atmosphere Monitoring Service (CAMS), with the dataset referred to as CAMS-GLOB-SOIL v2.2. These data are available through the Copernicus Atmosphere Data Store (ADS) system, (https://doi.org/10.24380/kz2r-fe18, last access June 2021, Simpson 2021a) or through the Emissions of atmospheric Compounds and Compilation of Ancillary Data (ECCAD) system (https://eccad.aeris-data.fr/, last access June 2021). For review purposes, ECCAD has set up an anonymous repository where a subset of the CAMS-GLOB-SOIL v2.2 data can be accessed directly
(https://eccad.aeris-data.fr/essd-surf-emis-cams-soil/, Last access July 2021, Simpson 2021b).

## 1 Introduction

This work presents a dataset of global soil emissions of nitrogen oxide (NO), designed for implementation in atmospheric chemical transport models. The dataset, denoted CAMS-GLOB-SOIL v2.2, is part of a family of emissions datasets intended to improve the representation of anthropogenic and biogenic emissions within the Copernicus Global and Regional emissions
service (CAMS81, Granier et al. 2019; Guevara et al. 2021; Kuenen et al. 2021; Sindelarova et al. 2021), directly supporting the CAMS production chains (https://atmosphere.copernicus.eu/).

Soil NO emissions are essential to regional modelling of e.g. ozone and particulate matter (PM), especially on the global scale, and a number of methodologies and datasets for these emissions of NO have been presented (Ganzeveld et al., 2002;



Steinkamp et al., 2009; Hudman et al., 2010; Visser et al., 2019). These methodologies have various levels of time-resolution,

sophistication and data-requirements, but typically rely on land-cover maps, estimates of nitrogen inputs (fertiliser, deposition) to the soils, and meteorological modifying factors such as temperature, soil water and/or precipitation.

As will be discussed further in Sect. 2, all estimates of soil NO emissions are rendered very uncertain due to a large number of factors associated with underlying datasets and also understanding of the source and sink processes of NO in soils. The factors which influence microbial activity, and NO production and loss are many and complex (Fowler et al., 2009; Butterbach-Bahl

et al., 2013; Skiba et al., 2020), and the underlying data (e.g. agricultural practices, soil textures, moisture) are difficult to assess (e.g Davidson and Kingerlee, 1997; Skiba et al., 1997, 2020; Veldkamp and Keller, 1997; Bouwman et al., 2002; Stehfest and Bouwman, 2006; Pilegaard, 2013). Therefore, an important component of this work has been to examine and elucidate the uncertainties in these emissions, and to seek a pragmatic merge of existing methodologies suitable for use in regional and chemical transport models. The emphasis has been on developing a framework with medium complexity, which does not rely

too much on complex and unverified data.

In future these emission estimates will be continuously improved through comparison with more complex models and with atmospheric concentration data (e.g. from satellites), which is becoming increasingly available.

As noted above, these NO emissions are intended for input to atmospheric chemical transport models (CTMs). Such CTMs are essential tools for the simulation and mapping of air pollution and radiative forcing, and for the design of effective emis-

sions abatement strategies. Within the CAMS system, the emissions produced in this study will be utilised within the CAMS Integrated Forecasting System coupled model (C-IFS Flemming et al., 2015; Wagner et al., 2021), but for the development and testing of the emissions we have made made heavy use of the model system of the EMEP MSC-W model (see Sect. 3.1).

In this article we will discuss firstly a brief summary of the processes controlling soil NO emissions (Sect. 2), the availability of relevant data (Sect. 3), and present the methodology (Sect. 4). Section 5 presents emission estimates and comparison with

other estimates. Sect. 6 discusses how the CAMS-GLOB-SOIL data can be used together with some key anthrogenic emission inventories, since some of the latter also include a soil-NO emission component (hence giving a risk of double-counting). Finally, Sect.7 presents some model simulations and comparison with measurements with and without the soil-NO emissions, in order to illustrate the impacts of these emissions on atmospheric $NO_2$ and $O_3$ concentrations. A brief overview of earlier versions of this inventory can be found in the Supplementary information.

The dataset is referred to as CAMS-GLOB-SOIL v2.2, with final calculations made in March 2021. Data are provided globally at $0.5° \times 0.5°$ degrees horizontal resolution, and with monthly time resolution over the period 2000-2018. Emissions are provided as total values and also with separate data for soil NO emissions from background biome values, and those induced by fertilisers/manure, pulsing and atmospheric deposition, so that users can provide their own modifications if wanted.

## 2  Nitrogen Oxide emissions: background

As discussed in e.g. Pilegaard (2013), the production and consumption of NO in soil is a result of both microbial activity and chemical reactions, and in general is controlled by four main factors: (a) N-inputs to the ecosystem, (b) temperature,

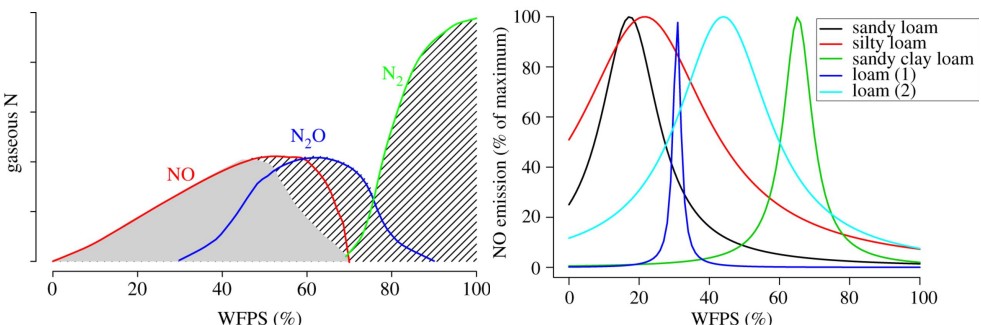

**Figure 1.** Dependence of NO emissions on soil water (here water-filled pore space, WFPS). a) (left) sketch (derived from Davidson et al. 2000) often used to illustrate the differing impact of WFPS on NO and $N_2O$ emissions; b) (right) Measured relationships between WFPS and NO emissions from chamber measurements from Schindlbacher et al. (2004). Figures reproduced with permission from Pilegaard (2013), where also more details are given.

(c) soil water content, and (d) soil pH. Many processes and microorganisms are involved; the two most important groups of microorganisms are nitrifiers and denitrifiers. Generally, both NO and $N_2O$ are produced by the same processes; however, the relative emissions depend on many factors that are not clearly understood. Many other factors play a role of course, with very complex relationships between microbial and chemical processes. (e.g Davidson and Kingerlee, 1997; Skiba et al., 1997, 2020; Veldkamp and Keller, 1997; Bouwman et al., 2002; Stehfest and Bouwman, 2006; Pilegaard, 2013). Even among forests, important differences have been found. For example, in a study of 15 forests sites as part of the European NOFRETETE project (Pilegaard et al., 2005), only a few of the coniferous forests were found to emit significant levels of NO, with deciduous forests or coniferous forests in low N-input areas having low emissions. NO emissions were higher in forest than grassland soils. The high NO emissions from forest soils were mainly attributed to low pH and high soil porosity, and NO emissions were positively correlated with N input.

Figure 1 provides an important example of the complexity of the soil-N system. Fig 1a illustrates the typical assumed response, whereby NO is emitted at low to moderate values of water-filled pore space (WFPS), here with maximum emissions at around 50-60% WFPS. Fig 1b however illustrates the findings of the relationships found in chamber studies by Schindlbacher et al. (2004). Clearly, different soils have very different characteristics, and so simple modelling systems cannot hope to capture such variability.

Global estimates of soil NO emissions have been available for many years. For example, Potter et al. (1996) used an ecosystem modeling approach (CASA) to integrate remote sensing, climate, vegetation and soil datasets. Monthly mean global distributions at $1° \times 1°$ grid resolution of the NO emission fluxes (and other gases) were produced. Kesik et al. (2005) and Butterbach-Bahl et al. (2009) produced European inventories of soil NO emissions from forests and grasslands using the biogeochemical models DNDC/Forest-DNDC system. In Butterbach-Bahl et al., DNDC captured differences in the magnitude of





NO emissions between sites, but was less successful when simulating observed day-by-day variations. However, major peak emission events, e.g. due to fertiliser application or following rainfall events, were mostly simulated. Results from another biogeochemical model, OCN (Zaehle et al., 2011), have been used in global runs of the EMEP MSC-W CTM (Simpson et al.,

2012; Jonson et al., 2018; Schwede et al., 2018). In this case the soil NO emissions are provided as monthly average values.

Methods of modelling short-term NO emissions for CTM applications have also been available for many years, though with recognised large uncertainties. The most widely used method in early efforts to include soil NO emissions was that of Novak and Pierce (1993), commonly known as the second version of the Biogenic Emissions Inventory System (BEIS-2). This method, derived from the results of Williams et al. (1992), has been applied previously in Europe by Simpson et al. (1995) and

Stohl et al. (1996).

An important paper with respect to global scale modelling was that of Yienger and Levy (1995), hereafter referred to as YL95. In this approach, emissions were parameterised as a function of biome type, temperature and precipitation. YL95 introduced a scheme to allow pulses of emissions when rain follows a dry spell, and they also introduced a canopy reduction factor (CRF) to allow for capture of soil-emitted NOx before escape from the canopy. Soil temperatures ($T_s$) were estimated

from air temperatures using simple empirical relationships.

The YL95 model has been widely used in atmospheric CTMs, because of its simplicity and link to readily available meteorological data (e.g. Ganzeveld et al., 2002). YL95 is also part of the MEGAN model (Guenther et al., 2006) which is already in use to provide BVOC emissions as part of CAMS-81 (Sindelarova et al., 2021).

The 'BDSNP' model of Hudman et al. (2012) can be seen as an update of the YL95 methods. The basic formulation

is similar, but soil moisture was handled with the use of water filled pore space (WFPS) values from a numerical weather prediction (NWP) model. The background emission factors from each biome ($A'_{biome}$) are similar to those used in YL95, but updated using data from Steinkamp and Lawrence 2011. The BDSNP model handles soil water in a smoother way than YL95, using a response curve similar to that shown in Fig. 1a. In BDSNP the optimum WFPS (denoted $\theta$ in Hudman et al. 2012) is given as 0.2 for arid soils and 0.3 elsewhere, with $\theta$ being provided by the top 2cm soil layer from GEOS-Chem meteorology.

Importantly, Hudman et al. noted that $\theta$ values had not yet been validated; an important caveat which was one of the reasons for the more simplified treatment of soil moisture which was applied in our CAMS-81 approach.

For N-inputs, Hudman et al. (2012) further used the fertiliser data of Potter et al. (2010), with fertiliser spread according to a Gaussian distribution around the green-up day of the crops. Data from satellites (MODIS and TRMM) were used to estimate growing seasons. A simple mass balance approach tracks the changes in soil $N_{avail}$ as a result of fertiliser inputs

and atmospheric N-deposition, with decay lifetimes of 4 months for fertiliser inputs and 6 months over landuse with natural vegetation. Rasool et al. (2016) extended this approach further using a more up-to-date and advanced treatment of fertiliser inputs from the agroecosystem EPIC model, which provided daily inputs over the U.S.A.

Dammers (2013) implemented this BDSNP model into the LOTOS-EUROS CTM (Schaap et al., 2008), and found that it lead to more realistic simulations of NOx concentrations. They had to adapt the LOTOS-EUROS land-cover classifications to

match the biomes of the BDSNP (Steinkamp and Lawrence 2011) system, also accounting for climate class. They also used soil temperatures from ECMWF's top 7cm layer, which is not quite the same as the 2cm layer used by Hudman et al. 2012,



but a logical option (also for CAMS simulations). Other details are given in Dammers (2013), who also show that soil-NOx emissions estimated using the BDSNP scheme are a significant contributor to summertime NOx emissions in some countries.

## 3 Data sets

One of the major problems in producing gridded and spatially disaggregated emissions of any natural/biogenic pollutant is lack of the necessary data. With soil emissions in particular, many variables are often considered important, but are almost never available in an accurate way. Among the most difficult variables are soil pH and redox potential, both of which can vary significantly within a model's grid-cell, and under different types of vegetation. Even variables that are nominally available to CTMs are often of very uncertain quality, and among these soil water (SW) stands out as very important for trace gas production
and emissions from soils. Agricultural management (including amount and timing of fertilizer applications) is also a crucial requirement for many emission models, but typically lacking or available only at the country scale (Stehfest and Bouwman, 2006).

For practical reasons, several of the data sets used in CAMS-GLOB-SOIL are taken either directly from the EMEP MSC-W modelling system, or derived from meteorology used in that system, so this is discussed first (Sect. 3.1), followed by brief
discussions of the other data-sets which are used for soil emission estimates. (Sects. 3.2-3.3.1).

### 3.1 The EMEP MSC-W model

The atmospheric CTM used to develop and test the NO emission system is that of EMEP MSC-W ( the Meteorological Synthesising Centre – West of the European Monitoring and Evaluation Programme). The EMEP MSC-W CTM (hereafter EMEP model) is a three-dimensional Eulerian model whose main aim is to support governments in their efforts to design effective
emissions control strategies (e.g. Simpson et al., 2012, 2020b, and references therein). The emission estimates presented in this paper were heavily based upon both input and output data from the EMEP model, since (a) the EMEP model's input meteorology is essentially the same ECMWF meteorology as used in the IFS model (Sect. 3.1), and (b) the EMEP model has been used to generate fields of N-deposition for the multi-year time-series used here.

Version 4.0 of the EMEP model was described in detail by Simpson et al. (2012), though substantial updates have been made
in the treatment of aerosols, biogenic emissions and chemistry over the years. These updates have been discussed in specific articles (e.g. Stadtler et al., 2018; Simpson et al., 2020a; Bergström et al., 2021) and in annual EMEP reports (Simpson et al., 2020b, and refs therein). The model has been used in several studies of N-deposition (Simpson et al., 2006a, b, 2014; Schwede et al., 2018; Tan et al., 2018; Theobald et al., 2019) and indeed soil N emissions (Kesik et al., 2005, 2006).

Although originally designed for European applications, the model is very flexible and is now applied on scales ranging
from global (Jonson et al., 2010, 2018; Schwede et al., 2018; McFiggans et al., 2019) to local (1-7 km grids) (e.g. Vieno et al., 2010, 2014; Schaap et al., 2015). In this work we use version 4.37 of the model, with $0.5° \times 0.5°$ resolution, as used for the Arctic Monitoring and Assessment Programme (AMAP, Whaley 2021).



## 3.2 Landcover

Biomes in v1.1 of CAMS-GLOB-SOIL were from the EMEP MSC-W landcover model's system which is a hybrid of the
GLC-2000 land-cover data-set (https://forobs.jrc.ec.europa.eu/products/glc2000/glc2000.php, last access June 2021), and the
Community Land Model (https://www.cesm.ucar.edu/models/clm/, last access June 2021), Oleson et al. 2010; Lawrence et al.
2011), as described in Simpson et al. (2017).

For v2.1 and v2.2, we have made use of a MODIS-based landcover, which corresponds closely to the landcover used by
SL11 in their analysis of emission factors. The data used were the MCD12C1v006 data set (Sulla-Menashe and Friedl, 2018;
Friedl and Sulla-Menashe, 2015), downloaded from https://lpdaac.usgs.gov/products/mcd12c1v006/#tools (last access June
2021). Of the available data sets, we used the LC Type 1 data, which give sub-pixel fractions of land-cover classified according
to the the International Geosphere-Biosphere Programme (IGBP) system.

This MCD12C1 data set provided 17 basic land-cover types. These were further disaggregated into the 23 categories of
SL11 by overlapping these data with the Köppen-Geiger climate classification as provided by Kottek et al. (2006). This new
landcover map allows direct application of the SL11 emission factors as detailed in Sect. 4.1 below. The resulting 23 land-cover
biomes are given in Table 2.

## 3.3 Meteorology

The main meteorological parameters needed for soil NO emissions are (a) soil moisture, and (b) soil temperatures. As noted
above, we have used meteorology from the European Centre for Medium Range Weather Forecasting Integrated Forecasting
System (ECMWF-IFS) model (http://www.ecmwf.int/research/ifsdocs/), as processed for the EMEP model in this work. These
data had 3-hourly time resolution, and a $0.5° \times 0.5°$ degree longitude-latitude grid.

### 3.3.1 Soil moisture index (SMI)

The IFS model and all NWP models provide the surface soil moisture in a uniform way and with good temporal resolution.
This seems the most practical solution for CAMS-81 in the initial stages at least, although some satellite-derived products may
provide alternatives at a later stage (e.g. Dorigo et al., 2017).

However, the accuracy of SW estimates from NWP models is questionable (e.g. Balsamo et al., 2009; Albergel et al., 2012;
Wipfler et al., 2011; Samaniego et al., 2013). There are also problems in converting between different soil water metrics, e.g.
from mass fractions to water filled pore space (WFPS). A further and important complication is that many agricultural areas
are subject to irrigation. Although data-sets such as GAEZ (Global AgroEcological Zones, https://www.gaez.iiasa.ac.at/, last
access June 2021) provide area fractions of irrigated versus non-irrigated crops, the timing and extent of irrigation is usually
unknown. These issues suggest that although it is reasonable to hope that NWP models get SW 'about-right', it would be
unreasonable to expect them to predict volumetric fractions, or (even more difficult) soil-water pressure, in an accurate way.
These data must be used pragmatically, and tested as part of the inventory process.



The EMEP model makes use of the so-called soil moisture index (SMI) which is available from the IFS model (ECMWF, 2021). Defining minimum and maximum soil water amounts to be the permanent wilting point (PWP) and field capacity (FC), SMI is defined as:

$$\text{SMI} = \frac{\text{SM} - \text{PWP}}{\text{FC} - \text{PWP}} \tag{1}$$

where SM is volumetric soil moisture, PWP is the permanent wilting point, and FC is the field capacity, all in $m^3\ m^{-3}$. SMI can be calculated in this way for each soil type in the grid, and then averaged to get a grid-average value which is more physically meaningful than a simple average over absolute volumetric soil moisture values. The SMI values used here ('SMI1') are from the upper 7 cm of the soil.

Although it is simply impossible to take into account all the variability associated with heterogenous vegetation and soil types within a grid-cell, this SMI index should hopefully capture the main episodes of soil drying and effects on vegetation.

### 3.3.2 Soil temperatures

Although the IFS model does provide soil temperatures, we have simply used 2m air temperature for the current calculations. There are two main reasons for this: (a) most importantly, this variable was easily available from the EMEP model system we were using, and (b) it is anyway difficult to interpret soil temperatures from a numerical weather prediction model in terms of ecosystem-specific values.

The latter point is important, as the relation between air and soil temperatures is complex, and depends upon the vegetation cover and its physiological state (e.g. LAI) over the year. Soil temperatures may be higher or lower than air temperatures, and the many parameters required may depend on topography, soil texture, and soil water content – all of which may vary over short distances and even over different types of crops (e.g. Zheng et al., 1993; Brown et al., 2000; Kang et al., 2000; Plauborg, 2002; Tsilingiridis and Papakostas, 2014).

### 3.4 N-inputs, Fertilizer

Nitrogen inputs to ecosystems are a main driver for most N-related emissions. In agricultural areas, fertilizer application is the main source of N (and sometimes nitrogen fixation). For semi-natural areas atmospheric N-deposition is a key input.

Maps of global fertilizer and manure inputs were estimated by Potter et al. (2010, 2011), for the period of around 2000-2007. These data were converted to maps of N availability with $0.5° \times 0.5°$ degrees spatial resolution and monthly time resolution for the HEMCO system (Keller et al., 2014, http://wiki.seas.harvard.edu/geos-chem/index.php/HEMCO, last acess June 2021). These data were derived from N-inputs spanning the years 2000-2007, but with most emissions for the latter year (Potter et al., 2010). Hence we assigned these data a nominal year of 2005.

Scaling factors to get to other years were made by combining national year to year variations from the CEDS database (Hoesly et al., 2018) with global $NH_3$ emission from ECLIPSEv5a database (https://iiasa.ac.at/web/home/research/researchPrograms/air/Global_emissions.html, last access June 2021) with the latter needed to allocate country codes to grids. For this first emis-



sion estimate, where we only attempt monthly resolution of emissions, we adopted the simple procedure of allowing emission rates to follow these monthly N-inputs.

### 3.5 N-inputs, atmospheric deposition

Estimates of atmospheric N-deposition are readily available from CTMs (Dentener et al., 2006; Simpson et al., 2014; Kanakidou et al., 2016; Schwede et al., 2018), though often for limited time periods or with coarser spatial resolutions than are used in
CAMS81. For this work, estimates of atmospheric N-deposition were taken from the EMEP chemical transport model (Simpson et al., 2012, 2020b), as run for the Arctic Monitoring and Assessment Programme (AMAP) project (Whaley, 2021). For these calculations a $0.5° \times 0.5°$ degrees resolution horizontal was used over the 2000-2015 period, building upon emissions from the ECLIPSE v6b dataset (https://iiasa.ac.at/web/home/research/researchPrograms/air/ECLIPSEv6.html, last accessed 24th March 2021). For the years 2017 and 2018 we have simply used the 2016 values.

It can be noted that there are large uncertainties in deposition estimates from all CTM models or indeed from observation-based estimates (Flechard et al., 2011; Schwede et al., 2011; Simpson et al., 2014; Vet et al., 2014; Theobald et al., 2019; Walker et al., 2020), but simple mass-balance should ensure that over the large scale the amounts deposited should be constrained by emissions.

### 4 Methods

The basic methodology merges methods from Yienger and Levy (1995) (hereafter YL95, c.f. Tab. 1), with various updates to reflect recent literature (especially Steinkamp and Lawrence, 2011, hereafter SL11), and some simplifications which reflect lack of availability of some key data. In YL95 and SL11, background biome emission factors ($A_{biome}$) were modified by estimates of locally available nitrogen ($N_{avail}$), which consists mainly of agricultural inputs of N (N from fertilizer, manure, hereafter $N_{Fert}$), or atmospheric deposition of reactive nitrogen (hereafter $N_{Dep}$), and a pulse factor, $F_{pulse}$. For this work we
prefer to calculate the contributions of $N_{Fert}$, $N_{Dep}$ and $F_{pulse}$ separately, so we have:

$$F_{soil} = F_{biome} + F_{Fert} + F_{Ndep} + F_{pulse} \tag{2}$$

where the flux terms have units ng(N) m$^{-2}$s$^{-1}$. The calculations of $F_{biome}$, $F_{Fert}$, $F_{Ndep}$ and $F_{pulse}$ are summarised in Sects. 4.1–4.4 below. Canopy-reduction factors (CRF) are discussed in Sect. 4.5. Issues associated with rainforests and estimation of soil temperatures are discussed in Sects. 4.6–4.7.

We have aimed at monthly resolution for this study. One important reason is that many of the underlying data-sets have monthly resolution, and even this has substantial uncertainties. Secondly, the most dramatic short-term variation with soil NO emissions is associated with pulses, and for reasons given in Sect. 4.4, estimation of the timing of such events cannot reliably be provided at this stage.

**Table 1.** Frequently used abbreviations

| | |
|---|---|
| SL11 | Steinkamp and Lawrence (2011) |
| YL95 | Yienger and Levy (1995) |
| v1.1 | CAMS-GLOB-SOIL v1.1, 2018 version of soil NO emissions. See Simpson (2018), Granier et al. (2019) |
| v2.1 | CAMS-GLOB-SOIL v2.1, Update of v1.1 |
| v2.2 | CAMS-GLOB-SOIL v2.2, This version of soil NO emissions. Update of v2.1 |

### 4.1 Calculation of $F_{biome}$

The basic emissions algorithm for $F_{biome}$ is given by:

$$F_{biome} = A_{biome} \times f(T, SMI) \times CRF \tag{3}$$

where $F_{biome}$ is the background biome-based soil NOx flux ($ng(N)\,m^{-2}s^{-1}$), $A_{biome}$ is a function of the biome-type, $f(T, SMI)$ is a function of temperature and soil moisture index, and CRF is the canopy reduction factor accounting for NOx-capture by the vegetation canopy above the soil.

The biome emissions, $F_{biome}$, are driven by the underlying land-cover data, biome factors ($A_{biome}$), and meteorological drivers. Following YL95 and SL11, biome factors are given for dry and wet soils, with different temperature functions ($f(T)$) used for both. With the updated landcover used in v2.1, values of the emission factors were now taken directly from SL11, as tabulated in Table 2.

As seen from Table 2, we need to distinguish 'dry' from 'wet' soils. YL95 defined soils as being dry when the accumulated 245 precipitation over the last 2 weeks was less than 1 cm, but subsequent authors have made use of NWP soil moisture data. SL11 defined the threshold between wet and dry soils at 15% volumetric soil moisture, which for an average soil was said to correspond to midway between PWP and FC, i.e. to SMI=0.5. Figure 2 illustrates the fraction of time that grid-cells are defined as wet with this SMI=0.5 threshold. Although not identical to the results shown in Fig.7 from Steinkamp and Lawrence (2011), the results are similar. We therefore define soils with SMI>0.5 as wet, otherwise dry.

As with YL95 and SL11, crops are assumed to be irrigated, and so the Aw rates applied at all times through the growing season. Defining this growing season is difficult for number of reasons though. This includes the wide variety of species, with different planting and phenological developments, and the possibility of multiple harvests in the same fields (e.g. Sacks et al., 2010; Mills et al., 2018). For this study we have made the simple assumption that the months in which fertilizer application rates (Sect. 4.2) are above the median values for any particular grid cell are those when crops are likely to be growing.



**Table 2.** Biome-based emission factors ($A_{biome}$, ng(N) m$^{-2}$s$^{-1}$) for dry and wet conditions, and canopy reduction factors (CRF), used for the MODIS/SL11 biomes used in this work.

|  | Biome[a] | $A_{biome}$[b] | | CRF |
|---|---|---|---|---|
|  |  | wet | dry | (fraction) |
| 00 | water | 0.00 | 0.00 | 0.00 |
| 01 | permanent wetlands | 0.00 | 0.00 | 0.50 |
| 02 | snow and ice | 0.00 | 0.00 | 0.00 |
| 03 | barren or sparsely vegetated DE | 0.00 | 0.00 | 0.00 |
| 04 | Unclassified | 0.00 | 0.00 | 0.00 |
| 05 | barren or sparsely vegetated ABC[c] | 0.00 | 0.00 | 0.00 |
| 06 | closed shrubland | 0.09 | 0.65 | 0.75 |
| 07 | open shrublands ABC | 0.09 | 0.65 | 0.75 |
| 08 | open shrublands DE | 0.01 | 0.05 | 0.75 |
| 09 | grasslands DE | 0.84 | 6.18 | 0.75 |
| 10 | savannas DE | 0.84 | 6.18 | 0.75 |
| 11 | savannas ABC | 0.24 | 1.76 | 0.75 |
| 12 | grasslands ABC | 0.42 | 3.07 | 0.75 |
| 13 | woody savannas | 0.62 | 5.28 | 0.75 |
| 14 | mixed forests | 0.03 | 0.25 | 0.50 |
| 15 | evergreen broadleaf forest CDE | 0.36 | 2.39 | 0.50 |
| 16 | deciduous broadleaf forest CDE | 0.36 | 2.39 | 0.50 |
| 17 | deciduous needleleaf forest | 0.35 | 2.35 | 0.50 |
| 18 | evergreen needleleaf forest | 1.66 | 12.18 | 0.50 |
| 19 | deciduous broadleaf forest AB | 0.08 | 0.62 | 0.50 |
| 20 | evergreen broadleaf forest AB | 0.44 | 2.47 | 0.30 |
| 21 | croplands | 0.57 | 0.00 | 0.75 |
| 22 | urban and built-up | 0.57 | 0.00 | 0.75 |
| 23 | cropland natural vegetation mosaic | 0.57 | 0.00 | 0.75 |

Notes: (a) labels such as 'ABC' denote the Köppen-Geiger categories associated with this biome, in this case A, B and C; (b) $A_{biome}$ factors are from Steinkamp and Lawrence (2011), except (c) values for barren or sparesely vegetated land-cover set to zero.



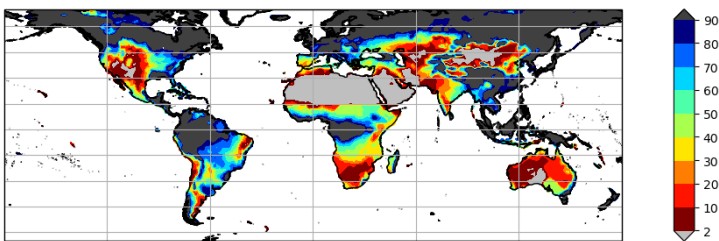

**Figure 2.** Percentage of wet soil conditions during 2012 given an SMI threshold of 0.5.

### 4.2 Calculation of $F_{Fert}$

Estimates of NO emissions from fertilizer N-inputs are commonly defined in terms of fertilizer induced emissions (FIE) - the percentage of the applied N which is released as NO. Steinkamp and Lawrence (2011) used FIE of 1%, but such estimates vary widely. YL95 used 2.5%, Bouwman et al. (2002) estimated 0.7%, and in an update of that work Stehfest and Bouwman (2006) used 0.55 for agriculture and grassland (excluding legumes). For v2.2 we have used an FIE value of 0.7%.

Potter et al. (2010) estimated N-inputs of 128.3 Tg(N) (relevant for the year 2007) through manures and 70.2 Tg(N) through fertilizers (relevant for the year 2000), giving 198.5 Tg(N). Assuming 0.7% of this is released as soil NO emissions, we estimate a contribution of just under 1.4 Tg(N) for the base-year of 2005 used in this work.

The final $F_{Fert}$ emissions for CAMS-GLOB-SOIL are then generated by applying this 0.7% factor to the annual global maps of N-inputs due to fertilizers and manures (Sect. 3.4). It should also be noted that these $F_{Fert}$ emissions emissions are sometimes, but not always, included in the agricultural sector of other emission data sets. The obvious risk of double-counting is addressed in Sect. 6.

### 4.3 Calculation of $F_{Ndep}$

As discussed in Sect. 3.1, N-inputs to soils from atmospheric deposition are estimated from monthly model results from the EMEP MSC-W chemical transport model. Emissions of NO are then estimated using the same re-release factor (0.7%) as used for fertiliser N-inputs. Given the large uncertainties in N-deposition estimates (e.g. Simpson et al., 2014) and relatively small contribution of the $F_{Ndep}$ term, this approach seemed acceptable for the current soil emissions calculation.

### 4.4 Calculation of $F_{pulse}$

Pulsing is the term used for the sudden emission of NO when soils that have been dry for some time are wetted. This release of NO is often of short duration. Both YL95 and SL11 used rainfall estimates in their approach to pulsing. In SL11 for example, if the accumulated precipitation was less then 10 mm in a gridcell during the last 14 days, and the precipitation then exceeds 1 mm ("sprinkle"), 5 mm ("shower") or 15 mm ("heavy rain") during one day, pulses of increasing magnitude and duration





(3-14 days) were triggered. Using this methodology, SL11, found pulsing fractions to be between 12–20% across all the land-covers (with mean value of 17%). The BDSNP model of Hudman et al. (2012) used soil water changes to initiate pulsing, but they also ackowledged that the soil moisture variable used ($\theta$) values had not yet been validated.

Although many studies suggest that pulsing is important, there is little evidence that such pulses can be accurately timed or quantified in global or even European scale CTMs. Indeed, Yan et al. (2005) noted that large scale NWP models have trouble predicting the conditions needed for pulsing, commenting that the ECMWF model's data never reached a value low enough to trigger a pulse in tropical savanna regions. Tests conducted for v1.1 showed that the timing of pulses varies greatly from one method to another (e.g. precipitation or SMI-based, and for different definitions of 'dry' versus 'wet'), so for v1.1 the pulsing

emissions were omitted.

As parameters such as volumetric soil water or the SMI used here cannot be verified, we have also explored some of the simpler rainfall-based approaches suggested in the literature. A very pragmatic methodology was devised for $F_{pulse}$ in v2.2. The occurrence of potential pulse events was counted using (i) a 14-day rainfall criteria (dry days were days with less than 1 mm rain per day, as long as SMI remained below 0.5), or (ii) changes in SMI of 0.01 after 3 days of SMI < 0.5 were counted. These

criteria in themselves often suggested quite different monthly distributions of possible pulsing events. Instead of choosing, both counts were simply summed, smoothed in time, and used as a normalising factor for the pulsing emissions. Firstly, the magnitude of annual emission was simply set to be 15% of the biome emissions set in Sect. 4.1 for each grid square where pulses were detected, loosely consistent with estimates by SL11.

Further work will be needed, for example based upon use of satellite soil moisture data and/or comparison to TROPOMI

NO$_2$ data (Veefkind et al., 2012), to find an algorithm which could be used with some confidence with regard to pulsing.

### 4.5  Calculation of CRF

It is well established that some of the NO emitted from soils can react quickly with ozone, forming NO$_2$. Some of this NO$_2$ is deposited within the canopy, reducing the emission of reactive N. YL95 used canopy reduction factors (CRFs) of between 0.25 for rain forests to 0.77 for Tundra, giving a global average of 0.53. These CRFs are very uncertain however, with Yan et al.

(2005) estimating 0.67 and Hudman et al. (2012) found 0.84. The CRF values used in this study, loosely based upon YL95 and Yan et al. (2005), are given in Table 2.

### 4.6  Tropical rainforests

The new land-cover data contains the category 'evergreen broadleaf forest' in Köppen-Geiger climates A&B, which was identified as 'rainforest' in SL11. As suggested by YL95, Steinkamp et al. (2009) and SL11, this tropical rainforest category

receives special treatment, in that the temperature functions are not applied, and instead dry/wet emissions are a function of season and not meteorology. Combined with the low CRF applies to rainforest the v2.1 and v2.2 emissions are then significantly reduced compared to v1.1 estimates. (We can note however that YL95 and SL11 differed greatly in the emission factors suggested for rain forests: YL95 suggested 8.6 and 2.6 ng(N) m$^{-2}$s$^{-1}$ for dry and wet soils respectively, whereas SL11 suggest just 2.47 and 0.44 ng(N) m$^{-2}$s$^{-1}$.)





According to Steinkamp et al. (2009), the dry and wet seasons of YL95 were defined in a very simple way, with 5 months of
      dry season each year, covering May–September in the Northern hemisphere and November–March in the Southern Hemisphere.
      Although this definition may suffice for annual calculations, this procedure leads to a large step change in emissions at the
      equator. For this work we have calculated the five driest months from a 5-year climatology of gridded rainfall. This procedure
      produces a much smoother transition in emissions changes near the equator. Having applied the dry and wet season emission

factors to this biome, we further apply a simple temporal smoothing to allow for the great uncertain in both the climatological
      shifts in emissions behaviour.

### 4.7   Temperatures

In S18, soil temperatures ($T_s$) were estimated from air temperatures using simple empirical relationships, $T_s(C) = T_a(C) + 5$
for dry soils (following YL95) and $T_s(K) = 0.72 T_a(K) + 82.28$ for wet soils (algorithm from the code base of the MEGAN

system, Guenther et al. 2012). However, closer examination of these equations, and alternatives as used by YL95 suggested
      by YL95, show some worrying features. For example, the MEGAN equations predict higher soil than air temperatures up to
      ca. 20°C, but in many situations this cannot happen, and indeed $T_a$ should often be higher than $T_s$. At 30°C temperatures the
      MEGAN system predicts $T_s$ of 27.4°C, whereas the Williams et al. (1992) equations used by YL95 would predict 28.9°C for
      grasslands and 28.8°C for forests - both close to air temperature. The ideal solution here would be to take $T_s$ from the ECMWF

model for each type of landcover, but this solution was not readily available for the current calculations. As an interim solution
      we simply assume that $T_s = T_a$, recognising that this needs to be improved in future methodologies.

## 5   Results: emission estimates

Figure 3 illustrates the calculated soil NO emissions for the year 2010, giving total emissions and the individual contributions
from $F_{biome}$, $F_{pulse}$, $F_{Fert}$ and $F_{Ndep}$. Time-series results for selected world regions (regions are shown in Fig. S1) are given

in Fig. 4, covering all years (2000–2018). These plots illustrate the strong spatial variations in soil NO emissions, and also
      that the drivers vary markedly from region to region. For example, western European emissions are estimated to be strongly
      affected by the fertilizer-induced emissions, whereas in southern Africa or South America it is the biome component that
      strongly dominates. Atmospheric deposition is seen to be a relatively small contributor, but of course the relative contribution
      will increase away from agricultural source areas. Overall, year-to-year variations are not especially large, and trends are rather

small.
      Month to month variations in emissions are much more prominent, as illustrated in Fig. 5. Seasonal cycles are driven largely
      by temperature and associated wet/dry changes. The large contribution of $F_{Fert}$ to Western/Eastern European(EUR) emissions
      is also very evident, with largest $F_{Fert}$ emissions near the start of the growing season.
      Finally, Fig. 6 illustrates the diurnal variations contained within the CAMS-GLOB-SOIL dataset for locations in Brazil,

France and Australia. These factors are derived from the temperature-dependent 3-hourly $F_{biome}$ biome values, but as a good
      first approximation they can probably be applied to the other components also. The main difference between the French and

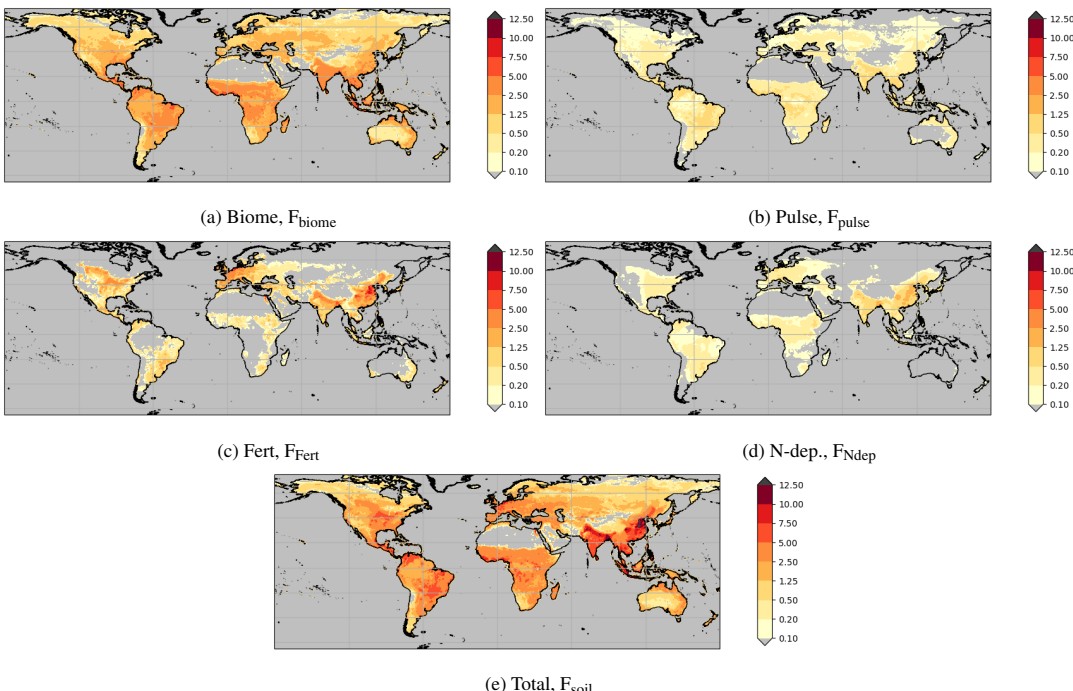

**Figure 3.** Above canopy NO emissions $(ng(N)\ m^{-2}s^{-1})$ calculated for year 2010: (a) Biome emissions (b) Pulse emissions (c) Fertilizer-induced emissions (d) Deposition-induced emissions (e) Total emissions

Australian examples given here is in the timing, since the dataset uses UTC times for all locations. The daily maximum factor is rather similar, at around 1.4. The factors for Brazil are different, and close to 1.0, a comsequence of the lack of temperature dependence in the tropical rainforest biome (Sect. 4.6).

**5.1 Comparison with other estimates**

Table 3 compares our estimates with other values from the literature, both globally and for some of the HTAP regions (c.f. Fig. S1). A valuable new data set in this regard is that of Weng et al. (2020), who used the HEMCO model (Keller et al., 2014) to calculate soil-NO emissions at $0.5°$ lat $\times\ 0.625°$ lon for 1980-2017 and $0.25°$ lat $\times\ 0.3125°$ lon for 2014-2017. The HEMCO algorithm is based upon the methods of Hudman et al. (2012), and is designed for use by models such as GEOS-Chem.

In general the global emissions fit rather well with literature values, including those of Weng et al. (2020). Estimates for Europe and southern Africa are almost identical. Estimates for North America are within 13%. Estimates for South America

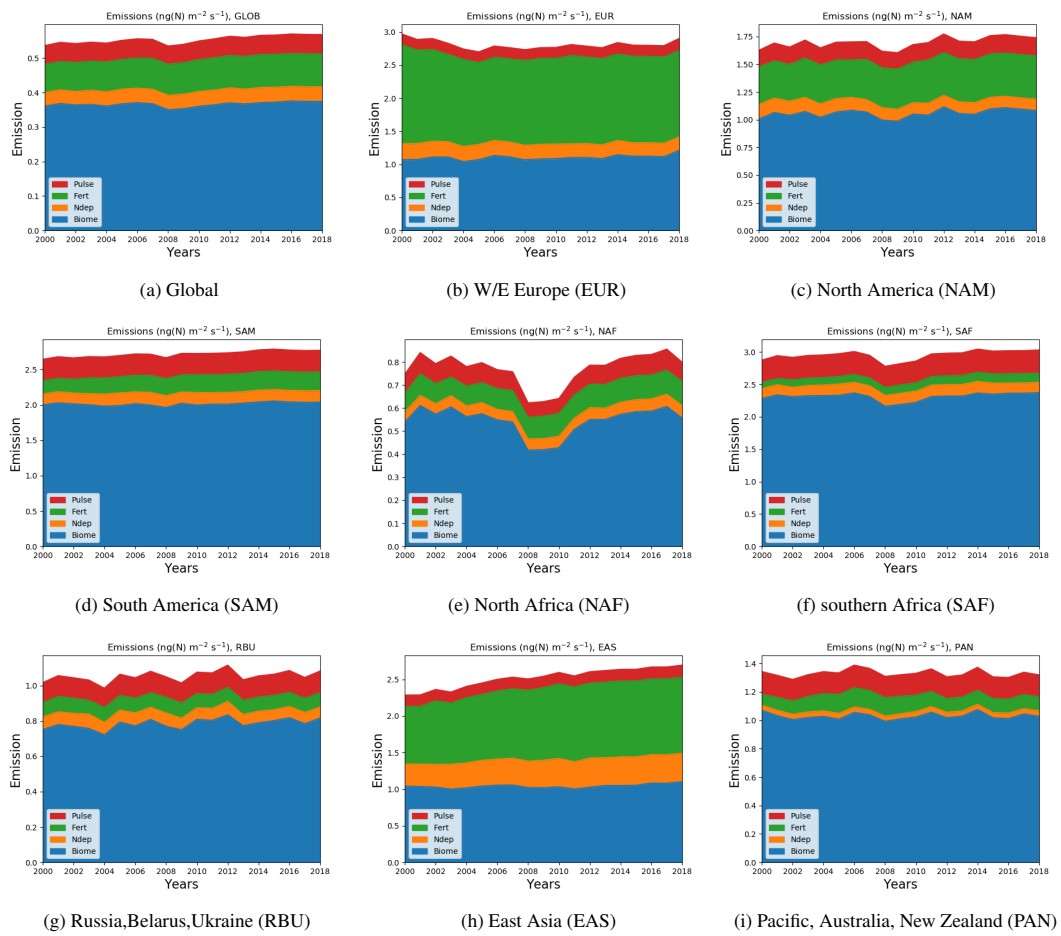

**Figure 4.** Above canopy NO emissions (ng(N) m$^{-2}$s$^{-1}$) calculated for years 2000-2018 for selected world regions. (Regions follow HTAP tier1 approach, c.f. Fig. S1.)



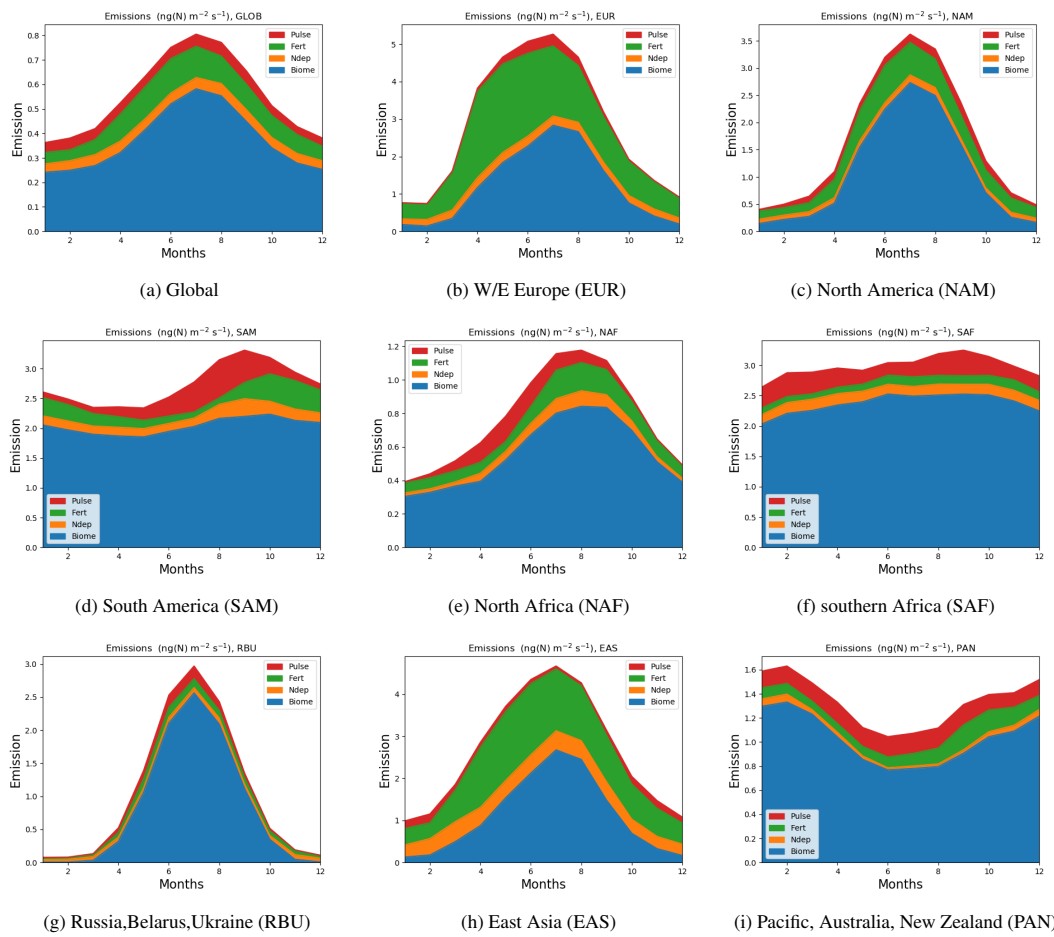

(a) Global      (b) W/E Europe (EUR)      (c) North America (NAM)

(d) South America (SAM)      (e) North Africa (NAF)      (f) southern Africa (SAF)

(g) Russia,Belarus,Ukraine (RBU)      (h) East Asia (EAS)      (i) Pacific, Australia, New Zealand (PAN)

**Figure 5.** Above canopy NO emissions (ng(N) m$^{-2}$s$^{-1}$) calculated for the year 2010 for the selected world regions.

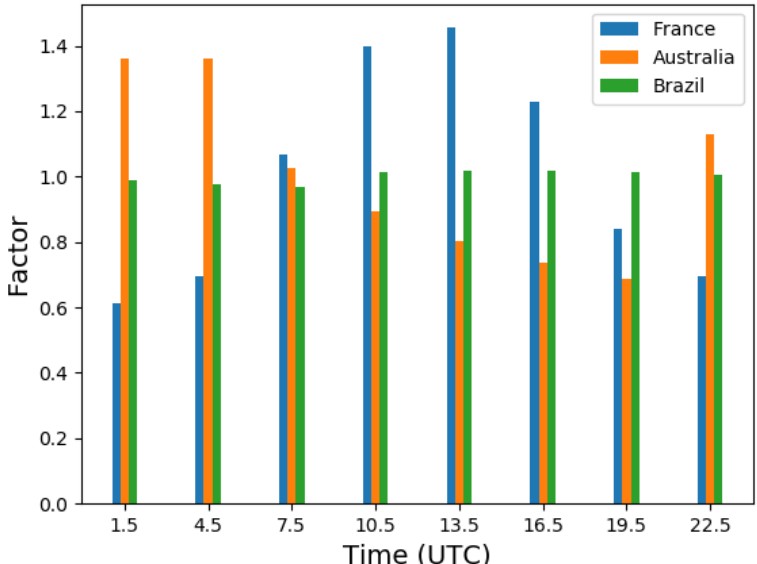

**Figure 6.** Diurnal variation factors for three locations: France (46°N, 3°E), Australia (28°S, 149°E), and Brazil (8°S, 46°E). Factors are given every 3 hours, centred on UTC times of 1.5, ... 22.5, for July 2015. (Bars are offset from these times for clarity.)

are 40% higher in v2.2 than in Weng et al. (2020), 53% for Russia, and 27% for East Asia. v2.2 emission estimates are substantially lower than Weng et al. (2020) for North Africa (21%) and South Asia (99%) The larger discrepancies for these regions probably reflects increasing difficulties with land-cover characteristics (e.g. savanna or sparsely vegatated areas) and 355 with the increasing frequency and importance of dry conditions.

The global satellite-based (OMI) estimate of Vinken et al. (2014). suggests somewhat larger global emissions than v2.2 or SL11, but the uncertainty range (±3.9 Tg(N)/yr) cited in that study is likely low since the analysis depends also on the use of a chemical transport model (GEOS-Chem) in the analysis.

## 6    Risks of doubling counting? Recommendations for different inventories

As discussed above, the CAMS-GLOB-SOIL v2.2 NOx inventory provides estimate of soil NO emissions from four categories, $F_{biome}$, $F_{Fert}$, $F_{Ndep}$, $F_{pulse}$, with the sum expressed as $F_{soil}$ (Eqn. 2). Given that the inventories to be discussed seem to include a





**Table 3.** Emissions (above canopy) of soil NO (Tg(N)/yr)

| Region | HTAP code | Emissions[a] | Sources |
|---|---|---|---|
| Globe | | 9.14 | v2.2, this study[a] |
| | | 8.8 | Weng et al. (2020)[b] |
| | | 12.9±3.9 | Vinken et al. (2014) |
| | | 9.0 | Hudman et al. (2012) |
| | | 11.6 | Zaehle et al. (2011) |
| | | 8.61 | Steinkamp and Lawrence (2011) |
| | | 4.97 | Yan et al. (2005) |
| | | 8.9 | Jaeglé et al. (2005) |
| | | 5.45 | Yienger and Levy (1995) |
| Europe | EUR | 0.48 | v2.2, this study |
| | | 0.47 | Weng et al. (2020) |
| | | 0.28 | Zaehle et al. (2011), for 2005 |
| | | 0.45 | Yan et al. (2005) |
| | | 0.11-0.7 [d] | Simpson et al. (1999) |
| Russia, Belarus, Ukraine | RBU | 0.47 | v2.2 |
| | | 0.26 | Weng et al. (2020) |
| North Africa | NAF | 0.32 | v2.2, this study |
| | | 0.75 | Weng et al. (2020) |
| | | 0.24 | Zaehle et al. (2011), for 2005 |
| southern Africa | SAF | 1.71 | v2.2, this study |
| | | 1.72 | Weng et al. (2020) |
| | | 3.24 | Zaehle et al. (2011), for 2005 |
| North America | NAM | 0.93 | v2.2, this study |
| | | 0.81 | Weng et al. (2020) |
| | | 0.63 | Zaehle et al. (2011), for 2005 |
| | | 0.64 | Yan et al. (2005) |
| South America | SAM | 1.34 | v2.2, this study |
| | | 0.84 | Weng et al. (2020) |
| | | 2.06 | Zaehle et al. (2011), for 2005 |
| | | 0.57 | Yan et al. (2005) |
| East Asia | EAS | 0.97 | v2.2, this study |
| | | 0.70 | Weng et al. (2020) |
| | | 0.72 | Zaehle et al. (2011), for 2005 |
| South Asia | SAS | 0.73 | v2.2, this study |
| | | 1.45 | Weng et al. (2020) |
| | | 1.47 | Zaehle et al. (2011), for 2005 |
| Pacific, Australia & New Zealand | PAN | 0.33 | v2.2, this study |
| | | 0.53 | Weng et al. (2020) |
| | | 0.24 | Zaehle et al. (2011), for 2005 |
| | | 0.46 | Yan et al. (2005) |

Notes: (a) HTAP domains (see Fig. S1) used to sum emissions from v2.2 and Weng et al.; (b) v2.2 values are averages over 2014-2017; (c) Weng et al. 2020 estimates are for 2014-2017 (0.25° lat × 0.3125° lon data), with regional sums over HTAP regions calculated from netcdf files to match CAMS estimates; (d) range is from estimates using 'Skiba' and BEIS-2 methodologies as applied by Simpson et al. 1999. The YL95 estimate was presented there as 0.6 Tg(N)/yr.



component similar to our $F_{Fert}$, we hereby introduce $F_{nonFert}$, such that:

$$F_{soil} = F_{nonFert} + F_{Fert} \tag{4}$$

where $F_{nonFert}$ is then the sum of the $F_{biome}$, $F_{Ndep}$, and $F_{pulse}$ terms.

Estimates of 'anthropogenic' soil NO are also provided by a number of emission inventories used by models from the CAMS system, including both the IFS model and EMEP MSC-W, and there are risks of both double-counting or omission of emissions when mixing CAMS-GLOB-SOIL with these other data sets. As will be shown below, many of the emissions derive their methods from the EMEP/EEA Emission Inventory Guidebook chapter on crop production and agricultural soils, so we present this first (Sect. 6.1), then for each emission data-set we present the status of soil-NO emissions, and a recommendation
on how these data should be used with CAMS-GLOB-SOIL.

### 6.1    Soil NO emissions in the EMEP/EEA Guidebook

Within the Convention on Long-range Transboundary Air Pollution (LRTAP Convention), most countries mainly report NOx emissions due to agricultural activities using the EMEP/EEA Emissions Inventory Guidebook (Hutchings et al., 2019). The Guidebook provides methods for calculating soil-NO data from fertilizer and other inputs.

Table S1 presents the main sources for which soil NO emissions ares covered by the Guidebook, and Table S2 presents the nationally submitted emissions following this system (data provided by Sabine Schindlbacher, EMEP CEIP, 2021). It can be seen that for the vast majority of countries the main emission categories are 3Da1,3Da2a-c,and 3Da3. These are all roughly equivalent to the 'Fert' emissions from CAMS-GLOB-SOIL.

### 6.2    WebDab/EMEP emissions

WebDab (https://www.ceip.at/webdab-emission-database) is the emission database of EMEP, and contains all emission data officially submitted to the Secretariat of the LRTAP Convention by the Parties to the Convention. When the detailed national emissions from webDab are compiled, gap-filled, and processed for use by the EMEP MSC-W, the WebDab 3D emissions noted above are included in the GNFR[1] category 'L' (emissions from agriculture 'other', which excludes livestock).

    Figure 7 clearly shows that the 'Fert' component of CAMS-GLOB-SOIL is remarkably similar to the sum ($3D_{tot}$) of the
WebDab '3D' categories components in most European countries. CAMS-GLOB-SOIL suggests much higher NO emissions for France, and also provides emissions for a few countries where soil NO emissions are lacking in WebDab (TR, UA, BY), but on the whole the agreement is good. It therefore seems reasonable to equate the 'Fert' emissions of CAMS-GLOB-SOIL with these GNFR L emissions as provided to EMEP MSC-W. This also suggests however that we need to add the nonFert emissions from CAMS-GLOB-SOIL to provide the best soil-NO estimate for modelling.


---

[1] GNFR=Gridding nomenclature for reporting/UNECE nomenclature for reporting of emissions to air, e.g. Matthews and Wankmueller 2020

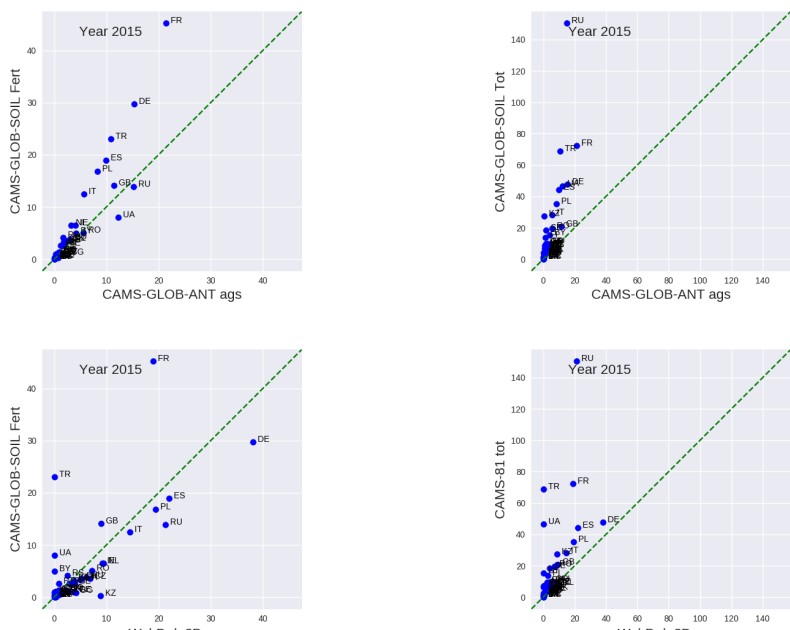

**Figure 7.** Comparison of CAMS-GLOB-SOIL emissions against CAMS-GLOB-ANT ags emissions and against National report emissions of category 3D from the WebDab system. For CAMS-GLOB-SOIL emissions are either from the 'Fert' category, or totals.

**Recommendation for EMEP/WebDab emissions:** *When using EMEP emissions derived from officially reported data (with soil NO emissions as given in GNFR L), for example in EMEP MSC-W reporting runs, retain the official GNFR L data, but add biome, N-dep and pulse emissions from CAMS-GLOB-SOIL.*

### 6.3 CAMS-REG

The anthropogenic European emissions provided by CAMS-REG (Kuenen et al., 2021; Granier et al., 2019) deliberately exclude soil-NO emissions, so as to avoid the risk of double-counting when used with CAMS-GLOB-SOIL (J. Kuenen, pers.comm., 2021). Thus, our recommendation is straightforward:

**Recommendation for CAMS-REG:** *Use GLOB-SOIL-NO directly when used with CAMS-REG.*




## 6.4 CAMS-GLOB-ANT, EDGAR

The CAMS-GLOB-ANT inventory (Granier et al., 2019) provides emissions from a number of emission sectors, with "agricultural soils" ("ags") as one specific category. The soil emissions from the ags source are derived from the EDGAR inventory (Crippa et al., 2018), which in turn uses the methods of the EMEP/EEA Guidebook (Monica Crippa, pers.comm.).

Figure 7 shows that the "Fert" component of CAMS-GLOB-SOIL is rather similar to the CAMS-GLOB-ANT components in most European countries, though CAMS-GLOB-SOIL provides somewhat higher emissions. The total emissions from CAMS-GLOB-SOIL can be significantly higher than those from CAMS-GLOB-ANT for some countries though, for example for Russia (RU) and Turkey (TR). In these cases a large land area provides for a large "Biome" component, and hence large national emission, which is not reflected in emission estimates which are based upon fertilizer applications only.

It is difficult to say whether CAMS-GLOB-SOIL-Fert is more realistic than CAMS-GLOB-ANT-ags even for the fertilizer-related emissions, since the results of both estimates show differences with the WebDab estimate. For example, for France the CAMS-GLOB-ANT-ags estimate of 21 Gg(N)/a is much closer to the WebDab estimate of 19 Gg(N)/a (CAMS-GLOB-SOIL-Fert suggests 45 Gg(N)/a), but CAMS-GLOB-ANT-ags suggests much lower emissions for Germany (DE, 15 Gg(N)/a) whereas CAMS-GLOB-SOIL-Fert suggests 30 Gg(N)/a and WebDab 38 Gn(N)/a. Further work is needed to resolve these

differences, but we can conclude:

**Recommendation for CAMS-GLOB-ANT, EDGAR:** *Use either:*

*i ags emissions from CAMS-GLOB-ANT (or EDGAR), plus biome, N-dep and pulse from CAMS-GLOB-SOIL. The ags*
*emissions currently have a flat seasonal cycle, however (Marc Guevara Vilardel, Barcelona Supercomputing Centre,*
*pers.comm., May 2021), which should be improved if such emissions are to be utilised.*

*ii  set ags emissions to zero, and use all CAMS-GLOB-SOIL emissions.*

Method [ii] should of course be the most consistent data-set, and both monthly and diurnal time-profiles are provided with the data-set, but more work to investigate the differences between the data-sets would be worthwhile.

### 6.4.1 ECLIPSE

The ECLIPSE inventories provided by IIASA (e.g. https://iiasa.ac.at/web/home/research/researchPrograms/air/Global_emissions.html, last access June 2021) have been widely used in global modelling studies (e.g. Stohl et al., 2013; Jonson et al., 2020). In earlier versions of the ECLIPSE inventory (ECLIPSE v5 and earlier) soil NO emissions were not included. In ECLIPSE v6b soil NO emissions are included, although in the same sector as other agricultural sources such as agricultural waste burning Chris Heyes and Z. Klimont, IIASA, pers.comm., 2021).

**Recommendation for ECLIPSE:** *Use GLOB-GLOB-SOIL directly when used with ELCIPSE v5 or ealier. Add Biome, Ndep and Pulse from CAMS-GLOB-SOIL when used with ECLIPSE v6.*



## 7    Modelling the impact of soil-NO

The CAMS-GLOB-SOIL estimates of soil-NO emissions are inherently difficult to validate, since (a) NOx emissions from other sources are ubiquitous, hence one cannot easily distinguish soil emissions from other sources, (b) there are few good-quality measurements of NOx in the rural areas where soil-NO emissions are expected to be important. The longer term goal is to make use of satellite data (e.g. OMI, Tropomi) to look for, evaluate, and calibrate the CAMS-GLOB-SOIL emissions, though this task is challenging for many reasons. As a first step towards emissions evaluation, and to get a better idea of the importance of soil NO emissions, we can however compare model runs with and without soil-NO emissions to measurements from well-established surface networks.

In this section we presents some preliminary calculations of the impacts of soil-NO. The EMEP model (v4.42) has been run for both the European domain (with resolution $0.2° \times 0.3°$ degree lat/lon resolution, and for the global domain with resolution $0.5° \times 0.5°$ degrees). For the European runs anthropogenic emissions are from the CAMS-REG dataset (Kuenen et al., 2021; Granier et al., 2019), and for global runs from CAMS-GLOB-ANT v5.1 (Granier et al., 2019), but with 'ags' emissions omitted to avoid double-counting as discussed in Sect. 6. The CAMS-GLOB-ANT dataset is based on the EDGARv5 annual emissions for the years 2000-2015 to which the monthly temporal profiles from CAMS-GLOB-TEMPO v2.1 (Guevara et al., 2021) have been applied. For 8 species including NO, ship emissions are from CAMS-GLOB-SHIP v2.1 (Jalkanen et al., 2012; Johansson et al., 2017). Emissions from aircraft are from CAMS-GLOB-AIR, and lightning and biogenic VOC are from EMEP model defaults (Simpson et al., 2012, 2017, 2020b).

Simulations are made with and without the CAMS-GLOB-SOIL emissions, for the years 2018 for the European run, and 2012 for the global run (consistent with Stadtler et al. 2018).

Figure 8 shows the increases in surface concentrations of $O_3$ and $NO_2$ from the global simulations. The changes in $O_3$ are significant, with around 1–2 ppb increases over most of Europe, and 2–4 ppb over much of North America, Asia and Oceania. Over much of sub-Saharan Africa and South America ozone increases by 4–8 ppb. Changes in $NO_2$ show a somewhat different spatial pattern to those of $O_3$, with the main hot-spots now in Asia. Changes in most other Northern hemispheric land-areas are typically of 0.2–0.6 ppb, with somewhat higher over Africa and South America. Over the oceans there is a belt of $NO_2$ decrease, but these changes are very small (usually less than 0.5 ppt), and presumably reflect increased $NO_2$ loss in the more chemically active troposphere induced by the soil NO emissions.

Table 4 summarises the evaluations statistics for the European run across a large number of stations, and for several gaseous and particulate compounds. Observations are from the EMEP network (Tørseth et al., 2012), and comprise stations in rural areas, suitable for evaluation of the EMEP CTM. Inclusion of soil NO emissions is seen to improve almost all statistics, with bias for N-compounds reduced significantly, but also correlation and IOA metrics are improved.

Table 5 summarises the evaluations statistics for ozone from the global run at a number of stations. Observations are from the GAW network (Schultz et al., 2015), and also comprise stations in rural and remote areas, suitable for evaluation of global CTMs. Here the results of adding soil NO emissions are seen to be more mixed. At the European stations we find similar responses to those discussed above, and especially improved R values at most sites (especially Payerne in Switzerland). For

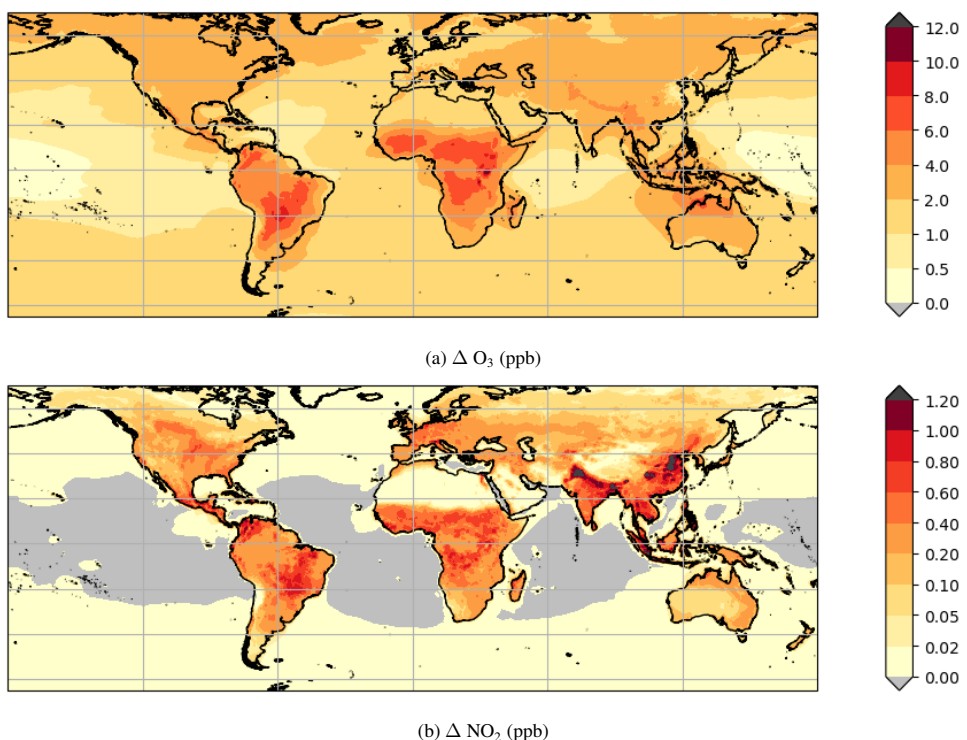

(a) Δ O$_3$ (ppb)

(b) Δ NO$_2$ (ppb)

**Figure 8.** Impact of soil NO emissions on surface concentrations of ozone and NO$_2$. Calculations with EMEP MSC-W model for 2012, using CAMS-GLOB-ANT emissions (minus "ags" sector) for anthropogenic emissions, and CAMS-GLOB-SOIL for soil NO emissions. See text for further details.

sites in N. America the soil-NO emissions sometimes lead to lower R-values (especially Trinidad Head in the USA, or Chapais in Canada). In Japan the R-values are relatively unchanged. Changes in other areas (Argentina, New Zealand, Cape Verde) are also relatively small.

The above comparisons are just meant as a quick snapshot of the impact of the CAMS-GLOB-SOIL emissions, and there is
a clear need to greatly expand the evaluation process. This will need to involve both a more detailed look at data from surface stations, and (probably most importantly) the use of satellite data.

**Table 4.** Comparison of modelled versus observed components over Europe, with and without (in parentheses) soil-NO emissions. Model results from EMEP MSC-W model (rv4.42, $0.2°\times0.3°$ degree lat/lon resolution, European domain), measurements from EMEP network (Tørseth et al., 2012).

| Component | Ns | obs | mod | bias (%) | RMSE | $R^2$ | IOA |
|---|---|---|---|---|---|---|---|
| Ozone daily max., ppb | 116 | 42.66 | 40.77 (39.95) | -4 (-6) | 3.49 (4.10) | 0.79 (0.77) | 0.81 (0.74) |
| NO in air, $\mu g(N)\,m^{-3}$ | 45 | 0.40 | 0.40 (0.36) | 0 (-11) | 0.29 (0.28) | 0.81 (0.80) | 0.89 (0.89) |
| $NO_2$ in air, $\mu g(N)\,m^{-3}$ | 73 | 1.71 | 1.73 (1.54) | 1 (-10) | 0.80 (0.80) | 0.84 (0.84) | 0.91 (0.91) |
| $HNO_3$ in air, $\mu g(N)\,m^{-3}$ | 17 | 0.12 | 0.09 (0.08) | -25 (-35) | 0.09 (0.10) | 0.44 (0.41) | 0.61 (0.58) |
| $NO_3^-$ in air, $\mu g(N)\,m^{-3}$ | 24 | 0.27 | 0.28 (0.25) | 4 (-7) | 0.11 (0.11) | 0.81 (0.81) | 0.90 (0.89) |
| $\sum HNO_3^+ NO_3^-$ in air, $\mu g(N)\,m^{-3}$ | 34 | 0.42 | 0.42 (0.37) | -1 (-14) | 0.08 (0.10) | 0.94 (0.94) | 0.97 (0.94) |
| $NO_3^-$ conc. in precip., mg(N)/l | 42 | 0.31 | 0.27 (0.23) | -12 (-24) | 0.24 (0.25) | 0.44 (0.43) | 0.56 (0.53) |

Notes: Statistics given are number of stations (Ns), observed and modelled values, bias, root mean square error (RMSE), correlation coefficient ($R^2$) and index of agreement (IOA).

## 8 Conclusions

We have presented a dataset of global soil NO emissions, CAMS-GLOB-SOIL v2.2, which comprises gridded monthly data, also with 3-hourly weight factors, suitable for atmospheric chemistry modelling. Data are provided globally at $0.5°\times0.5°$

degrees horizontal resolution, and with monthly time resolution over the period 2000-2018. Emissions are provided as total values and also with separate data for soil NO emissions induced by fertilizers/manure, pulsing effects, and atmospheric deposition, so that users can include, exclude or modify each component if wanted.

It should be emphasised that all estimates of soil NO emissions are notoriously uncertain, since the emissions are driven by complex under-soil processes (microbial activity, pH, organic-C content, nutrients) rather than the simple meteorological and

air quality variables which CTMs usually deal with, and there are very few data which can be used to evaluate such estimates. For example, Davidson et al. (2000) suggested that although their review of data (covering many tropical ecosystems) clearly supported the assertion that nitrogen oxide emissions are related to rates of nitrogen cycling in ecosystems, a model based on these regression parameters will have only order-of-magnitude prediction accuracy. Further, the emissions can vary markedly with vegetation type, fertilizer type and agricultural management systems, and prior occurrence of biomass-burning (e.g. Skiba

et al., 1997; Bouwman et al., 2002; Steinkamp and Lawrence, 2011).

For the CAMS-GLOB-SOIL datasets, we have here aimed at pragmatic solutions rather than sophistication, in order to set up a transparent initial framework, and to avoid over-parameterising a model in which many of the underlying datasets (e.g. on agricultural inputs, or soil characteristics) are necessarily uncertain. For example, the implementation of pulsing as done here (essentially, as 15% of the biome-related emissions where some moisture criteria was met) is deliberately simple, and



**Table 5.** Comparison of modelled versus observed means of daily maximum $O_3$ (ppb) at global GAW sites with and without (in parentheses) soil-NO emissions. Model results from EMEP MSC-W model (rv4.42, $0.5° \times 0.5°$ degree lat/lon resolution, global domain), measurements from GAW network (Schultz et al., 2015).

| Site | Country | LatN | LonE | Alt | DC (%) | Obs | Mod | Bias (%) | R |
|---|---|---|---|---|---|---|---|---|---|
| Ushuaia | Argentina | -54.8 | -68.3 | 18 | 97 | 24.8 | 23.9 (22.8) | -3 (-7) | 0.79 (0.77) |
| Kejimkujik | Canada | 44.4 | -65.2 | 127 | 97 | 37.1 | 40.1 (38.0) | 8 (2) | 0.51 (0.55) |
| Algoma | Canada | 47.0 | -84.4 | 411 | 98 | 40.1 | 41.0 (37.7) | 2 (-5) | 0.64 (0.64) |
| Saturna | Canada | 48.8 | -123.1 | 178 | 99 | 37.1 | 36.2 (34.9) | -1 (-5) | 0.71 (0.71) |
| Experimental Lakes | Canada | 49.7 | -93.7 | 369 | 99 | 37.9 | 38.1 (34.3) | 1 (-8) | 0.68 (0.70) |
| Chapais | Canada | 49.8 | -75.0 | 381 | 100 | 36.6 | 35.4 (32.4) | -2 (-10) | 0.72 (0.80) |
| Trinidad Head | USA | 41.0 | -124.2 | 120 | 92 | 35.5 | 36.3 (34.3) | 2 (-2) | 0.56 (0.66) |
| Cape Verde Obs. | Cape Verde | 16.8 | -24.9 | 10 | 99 | 35.0 | 39.6 (38.4) | 13 (10) | 0.90 (0.89) |
| Waldhof | Germany | 52.8 | 10.8 | 74 | 99 | 38.7 | 38.0 (36.1) | -1 (-6) | 0.86 (0.83) |
| Neuglobsow | Germany | 53.2 | 13.0 | 65 | 98 | 36.5 | 37.8 (35.9) | 4 (-1) | 0.81 (0.81) |
| Zingst | Germany | 54.4 | 12.7 | 1 | 99 | 38.1 | 38.5 (36.7) | 1 (-3) | 0.80 (0.78) |
| Westerland | Germany | 54.9 | 8.3 | 12 | 96 | 41.8 | 40.3 (38.6) | -3 (-7) | 0.80 (0.78) |
| Mace Head | Ireland | 53.3 | -9.9 | 8 | 100 | 41.7 | 40.2 (38.1) | -3 (-8) | 0.64 (0.66) |
| Zoseni | Latvia | 57.1 | 25.5 | 182 | 98 | 46.0 | 35.8 (33.8( | -21 (-26) | 0.75 (0.78) |
| Giordan Lighthouse | Malta | 36.1 | 14.2 | 160 | 94 | 47.9 | 48.9 (47.5) | 2 (0) | 0.70 (0.69) |
| Kollumerwaard | Netherlands | 53.3 | 6.3 | 0 | 95 | 37.2 | 37.9 (36.5) | 2 (-1) | 0.77 (0.75) |
| Vindeln | Sweden | 64.2 | 19.8 | 271 | 95 | 35.0 | 33.1 (33.1) | -5 (-10) | 0.75 (0.79) |
| Payerne | Switzerland | 46.8 | 7.0 | 490 | 99 | 41.1 | 42.8 (40.7) | 4 (0) | 0.74 (0.66) |
| Minamitorishima | Japan | 24.3 | 154.0 | 8 | 87 | 33.2 | 35.1 (34.3) | 6 (3) | 0.87 (0.86) |
| Yonagunijima | Japan | 24.5 | 123.0 | 30 | 98 | 44.9 | 53.2 (52.1) | 18 (16) | 0.75 (0.74) |
| Tsukuba | Japan | 36.0 | 140.1 | 25 | 99 | 46.1 | 49.5 (48.7) | 8 (6) | 0.72 (0.72) |
| Ryori | Japan | 39.0 | 141.8 | 260 | 98 | 46.6 | 45.2 (43.9) | -2 (-5) | 0.69 (0.68) |
| Lauder | New Zealand | -45.0 | 169.7 | 370 | 92 | 24.6 | 26.8 (24.8) | 9 (1) | 0.59 (0.6) |

Notes: Statistics are data capture (DC), Observer and Modelled daily max $O_3$, bias, and Correlation Coefficient (R).

reflects difficulties in even identifying the timing of pulses, let alone the magnitude. There are also some puzzling differences in the emission rates assigned to different land-cover by SL11, e.g. that the rates for mixed forest are lower than those of any deciduous or coniferous forest (cf Table 2). These differences presumably reflect a lack of measurement data, and this is a fundamental problem.

Future revisions to this data-set will hopefully include improved estimation of soil temperatures, inclusion of the impact of forest-fires, and generally more use of field data and satellite products to evaluate and constrain the estimated emissions.

## 9 Data availability

These data are available through the Copernicus Atmosphere Data Store (ADS) system, (https://doi.org/10.24380/kz2r-fe18, last access June 2021, Simpson 2021a) or through the Emissions of atmospheric Compounds and Compilation of Ancillary Data (ECCAD) system (https://eccad.aeris-data.fr/, last access June 2021). For review purposes, ECCAD has set up an anony-



mous repository where a subset of the CAMS-GLOB-SOIL v2.2 data can be accessed directly (https://eccad.aeris-data.fr/essd-surf-emis-cams-soil/, Last access July 2021, Simpson 2021b).

*Author contributions.* DS developed the soil NO emissions algorithms and generated the emission data, as well as writing most of this paper. SD helped with conversion of MODIS land-cover data and with provision of the anthropogenic emissions used in the modelling, as well as with many technical issues associated with the ECCAD database.

*Acknowledgements.* The presented work was supported by project CAMS_81: Global and Regional Emissions funded within the Copernicus Atmosphere Monitoring Service (CAMS, https://atmosphere.copernicus.eu/), coordinated by Claire Granier of the Centre National de la Recherche Scientifique (CNRS, France) and (from 2021) by Hugo Denier van der Gon (TNO, The Netherlands). The Copernicus Atmosphere Monitoring Service (CAMS, https://atmosphere.copernicus.eu/) is operated by the European Centre for Medium-Range Weather Forecasts on behalf of the European Commission as part of the Copernicus Programme. Additional funding was provided by EMEP under UNECE.

Thanks are also due to Bram Maasakkers (GEOS-Chem Support) for help in interpreting the HEMCO implementation of the Potter N-input datasets, Barron Henderson (US EPA) for spotting a bug (oceanic emissions) and for helpful discussions, and Meiyun Lin (US NOAA) for bringing up the issue of potential double-counting between CAMS-GLOB-ANT and CAMS-GLOB-SOIL. Ute Skiba (CEH, Scotland), Nick Hutchins (Århus University, Denmark) J. Webb (Univ. Wolverhampton, UK), Claire Granier (Laboratoire d'Aérologie, France), Jeroen Kuenen (TNO, Netherlands) and Sabine Schindlbacher (EMEP CEIP, Austria) are thanked for help in interpreting the definitions of soil NO

emissions with respect to the EMEP/EEA Emission Inventory Guidebook. Michael Gauss (MET Norway) is thanked for help in accessing atmospheric deposition data as provided to the AMAP project.



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
