# Peer review of "Global soil NO emissions for Atmospheric Chemical Transport Modelling: CAMS-GLOB-SOIL v2.2"

_Earth System Science Data, 2021_

## Referee Comment (RC4)

ESSD-2021-221

Authors provide relevant recipe  and guide for use of NO-forced transport models to better predict ozone or N2O. But, basically this conclusion: complicated combination of microbial and chemical processes, can never get key forcing functions (e.g soil temperature, soil moisture)j as we might desire, therefore adopt a pragmatic approach for modeling with perhaps (unverified) reasonable impact. Skillful? From a modeling perspective but does not really introduce new data nor address data quality (quantitative uncertainty of model products) issues as expected for ESSD. Submit elsewhere?

Line 154: Acronym SL11 used here prior to its formal definition which occurs later at line 221 and again at line 234 in Table 1.

Skillful use of data from many sources. All sources listed (by URL with most recent access) in manuscript text. For this reader/user, a table of sources, with reference (if available) or contact information plus existing URL would prove very useful. Perhaps as an appendix? Perhaps in order of presentation as opposed to e.g. alphabetical? Something like Table 1 but for web-based data products used here?

Line 232: Implies that variability in NO emissions resulting from so-called 'pulses' dominate overall variability in NO emissions but, as this reader understands and as authors have repeatedly noted, periodic N-fertilizer inputs have equal or greater impact?

Line 254: Needs clarification of units. Percentage of 2012 max SMI for land type? Of monthly max? Of growing season max? Why apparent floor of 2%?

Line 260: here NO emissions represent only 0.7% of fertilizer N inputs but still account for nearly 1.4 Tg? A few lines later, around line 270, authors describe atmospheric deposition as highly variable but as "relatively small contribution". Conclusion here seems counter to that which occurred around line 232 (prior comment). Dealing with variability in one case vs absolute amounts of N inputs in another? If not clear to this reviewer, also not clear to readers?

Line 273; pulses defined here as "sudden emission of NO when soils that have been dry for some time are wetted". Looking back at equation 2 we get integrated soil flux (emission) of NO from a combination of emissions driven background (biome) inputs, by fertilizer inputs (disturbances), as outcome of atmospheric deposition inputs (steady or periodic?), and by so-called pulses which closely relate to episodic (precip-driven) changes (drying or wetting?) of soil moisture (technically, WFPS). Figure 1 implies NO emissions at lower WFPS followed by N2O emissions at higher WFPS, albeit with NO emissions highly dependent on soil composition (Fig 1b). If pulses so determinant to NO emissions, why do they seem so ill-defined? Weakness of manuscript presentation or ignorance of reader?

Line 277 and following: Pulsing (defined here as moisture increases above some minimum, but never as below some maximum) fractions as 12 to 20% OF WHAT? In following sentence, mean value of 17% OF WHAT? Further confusion around the 'pulsing' term? Troublesome, as - by authors' admittance - no verification possible? Further work needed to build confidence? Indeed!

Line 299: rain forest needs no capitalization but Tundra does?

Line 315: great 'uncertainty' rather than great 'uncertain'? This reader understands some reasons for treating topical forests in separate moisture-driven rather than temperature-driven scenarios, but in fact we have no idea how vegetation types or phenologies at any latitude or in

any biome category work annually in terms of NO emissions? Much less inter-annually? Much much less in any manner amenable to reliable parameterization?

Line 341, Figure 3: To my eye, biome and fertilzer have significant impacts, atmospheric deposition and pulsing much less so. Text so far would have lead me to a different conclusion? Same for most regions and most years in Figure 4 and for most regions and most seasons in Figure 5?

Line 352, Figure 6: inclusion of error estimates (e.g. 95% uncertainties) would render this figure entirely invalid?

Line 357: cite a 'low' uncertainty of a prior study (Vinken et al.) as reason to discount that work while failing entirely to summarize a cumulative uncertainty for this work?

Line 363: By this point authors have lumped fertilized vs non-fertilized. Do we now get uncertainties of each category? Or justification of how unquantified (but probably high) uncertainties have forced re-assignment of terms? Evidence presented so far suggests they should not lump biome with deposition and pulses?

Sections 6.1 to 6.4: Text here mostly relates to which sources to include for modeling purposes. Recommendations address only the double-counting issue, never any uncertainty issues (despite author-reported similarities or divergences in scatter plots of Figure 7, all sans uncertainties).

Line 466 Figure 8 and related text (Section 7): Figure 8 provides no uncertainty metrics, therefore related text (e.g. line 452) about O3 changes of 1-2 ppb as significant have in fact no validity for actual measurements. To the eye of this reader, the fact that most of northern hemisphere in Fig 8a shows, apparently, increases of 2 to 4 ppb O3 indicates background, e.g. that the model can't distinguish anything but broad equatorial increases compared to uniform insignificant background. Perhaps signal-to-noise somewhat better for N2O (Fig 8b) but again without an indication of sensitivity / errors. Because the model depends - as authors repeatedly remind us - on accurate soil moisture - why does Fig 8 show any values over ocean? Figure 8 has neither validity nor utility to this reader! From line 470: "clear need to greatly expand the evaluation process." Indeed!

For nearly every NO emission parameter authors correctly describe an ideal before describing their pragmatic solution in the meanwhile. Good! Necessary! But an encompassing uncertainty analysis should include and account for all of these necessary adjustments and assumptions? Authors approach uncertainties tangentially in their intercomparisions to existing products (e.g. Section 6) but this reader resonates with their conclusion "notoriously uncertain". A rigorous uncertainty analysis, e.g. for each term in Equation 2, would show most assumptions invalid at best. Understand that they (and we) need to do the best we can to understand and model NO emissions and consequent impact on ozone and N2O but here one encounters only a highly-pragmatic (necessary) approach lacking expected quantitative uncertainties and cautions. "notoriously uncertain" = not yet quantifiable and, as a consequence, not yet ready for publication? Starting again from Equation 2, which terms have uncertainties so large one needs to still discount them entirely? In conclusions: what do we know, what remains uncertain, what do we need to improve to fix weaknesses? Authors set up the question and evaluate modeling approaches but fail to resolve fundamental data problem.

Line 745: Why does SL11 reference (Steinkamp & Lawrence) include both DOI and ACP url?

---

## Author Comment (AC1)

**1 Reply to Referee #1**

We are replying quickly to Referee #1's comments as we disagree with the criticism that this manuscript and the dataset lacks sufficient innovation to be acceptable for publication, and believe that it is important to clarify the nature of the manuscript and of the CAMS-GLOB-SOIL v2.2 database. We strongly believe that datasets used by Copernicus, EMEP or indeed any Earth System modellers need to be documented, and we believe ESSD is the appropriate journal for such documentation.

**General comments:**

**Ref#1** This study implemented the YL95 soil NO emissions scheme with various

updates from other publications in the EMEP MSC-W chemistry transport model, and generated global soil NO emissions at the spatial resolution of 0.5 degree by 0.5 degree during 2000-2018. Soil NO is a significant contributor to global NO emissions, and generating global NO emissions inventory is important. However, it is very difficult to find innovation in this study, thus I cannot recommend its publication in ESSD.

**Reply:**

*We were disappointed to read the referees comments, but disagree with them, especially those related to 'innovation', and the need to publish manuscripts documenting datasets which are in open-source repositories. This paper was submitted to the journal 'Earth System Science Data', which we believe is a journal intended to describe datasets which are used in Earth System and similar models. The dataset we are describing, CAMS-GLOB-SOIL v2.2, is already available online, with a doi as required by ESSD. The CAMS-GLOB-SOIL data are in use already within both the Copernicus CAMS projects and the Air Convention's EMEP projects. Given this usage, we feel it is very important that the dataset is described in a publication, and that the main purpose of such a publication is to describe the methods and some consequences of the use of the data.*

*The referee argues that it is difficult to find innovation. As discussed below, we think that there are several innovative features. In any case it is important to note that this work was funded initially by the Copernicus CAMS81 project, and the need was for a practical dataset that improved upon the data initially used*

*in CAMS, and which could form the basis for future improvements. As with all CAMS81 datasets (e.g. for anthropogenic emissions, biogenic VOC), the soil NO methods build upon previously existing knowledge and data-sets, but modify them in an effort to better suit the needs of the CAMS modelling work. Such needs include:*

- *Datasets that can be kept up-to-date, and updated quickly. (We will soon calculate emissions for 2019 and 2020 with the same system, and this will continue as long as CAMS and/or EMEP have need of new data.)*

- *Datasets that can be extended to past and future scenarios: modelling of scenario years such as 2050 or historical years such as 1900 need to be possible, often with limited resources provided to adapt emissions to such years.*

- *Datasets which use meteorology from ECMWF, to be consistent as far as possible with the CAMS IFS model.*

- *Datasets that can be continuously tested within the CAMS community, and modified as a result of feedback and requests from different modelling and observation groups. This requirement by itself is a major reason why the CAMS-GLOB-SOIL dataset was developed inside the CAMS project, and not just taken "off the shelf" from some external source.*

- *Datasets that provide more detail than 'traditional' soil-NO emissions, especially with the contributions of fertilizer-induced emissions clearly delineated.*

*The referee suggests (below) that since Weng et al 2020 have published emissions for 2000-2018, then there is no need for another inventory. Actually the work which lies behind CAMS-GLOB-SOIL was started in 2018, and the basic methods built up well before Weng et al 2020 was published. We believe there is room for more than one soil-NO emission dataset in the atmospheric sciences community. Indeed, given the massive uncertainties associated with soil-NO emissions we believe that alternative datasets are very much needed in order to provide some indication of these uncertainties.*

**Specific comments:**

**Ref#1**

This study just implemented an old NO emissions scheme with various updates from other publications, but it lacks innovation. Moreover, Weng et al. (2020) has already generated global soil NO emissions at the resolution of 0.5 degree by 0.625 degree for 1980-2017. What is the innovation of this study compared with Weng et al. (2020). A little finer resolution is not sufficient to make it publish.

**Reply:**

*The referee further claims that we have implemented 'an old emission scheme' (YL95) with various datasets. This is misleading, as the same statement could be made about the Hudman et al 2012 inventory, and indeed the HEMCO emissions as used in Weng et al. (2020). These inventories all rely on the ideas introduced in YL95 of biome-based emission factors, which are modified by environmental and nitrogen-input factors. The Hudman type improvements (especially the use of smooth functions to represent the soil-water effects) do look very elegant, and we hope to introduce something like that in future versions of CAMS-GLOB-SOIL. However, though Hudman et al stressed themselves that the soil water itself could not be validated (we mention this on line 100). This type of uncertainty was one of the driving factors to our approach to use 'pragmatic' solutions for CAMS-GLOB-SOIL.*

*It is anyway more true to say that we start with the Steinkamp and Lawrence 2011 (SL11) emissions, which updated YL95 on the basis of 100s of new measurements. We have then developed simplified methods of dealing with fertilizer- and deposition- induced emissions, soil water effects and pulsing. Compared to most other papers on soil-NO emissions, we stress throughout the difficulties associated with both the input data and the methods used, explaining the pragmatic choices that were made.*

1. *The CAMS-GLOB-SOIL database is (as far as we know) the only dataset in which the contributions from background biome, fertilizer, deposition and pulsing are provided separately:*

$$F_{soil} = F_{biome} + F_{Fert} + F_{Ndep} + F_{pulse} \tag{1}$$

*Or*

$$F_{soil} = F_{nonFert} + F_{Fert} \qquad (2)$$

*This feature is both innovative, and very important! As we make clear in Sect. 6 there are substantial risks of double-counting when soil NO emission databases are used together with other emission inventories. Sect. 6 explains the problems, and suggests solutions, for dealing with this double-counting risk.*

*The system is proving extremely important in both CAMS and EMEP model runs, where users have to sometimes include and sometimes exclude the fertilizer component. This separation also provides a useful system whereby the different components can be modified by users. This allows for example users to specify different assumptions about the percentages of N released from fertilizers or N-deposition.*

2. *As part of this study we have also documented the usage of soil-NO emissions in EMEP, ECLIPSE, EDGAR and CAMS inventories of 'anthropogenic' emissions. Actually before this comparison was done, many colleagues were not aware of the extent to which soil-NO emissions had been included in these other inventories. We hope that the current paper will give guidance to many atmospheric modellers, even if they use other soil NO emission datasets.*

3. *The CAMS-GLOB-SOIL data use different meteorological and land-cover inputs to the Weng et al data. As the CAMS data are intended to be used with the CAMS Integrated Forecasting System coupled model (C-IFS), it is important that the meteorology is as consistent as possible with C-IFS. (In future the land-cover may also be changed in CAMS-GLOB-SOIL, C-IFS, and indeed other CAMS components such as forest-fire or biogenic VOC calculations to improve harmonisation. Discussions have started on such issues, but this is likely a multi-year process. The fact that such discussion exists among CAMS partners is however a good reason for CAMS to have control of its own soil emission methods.)*

4. *Sect.7 illustrates the results of applying CAMS-GLOB-SOIL at both global and regional scale. The literature has surprisingly few examples which show the impact of such emissions on atmospheric pollutants, either in magnitude or in comparison with observations. Although we restricted the number of results we used since this is a data-description and not a modelling*

*paper, we believe the snapshot we present of the impacts of soil NO can be useful for other scientists. (More detailed comparison with measurements, including satellite data, will be an important component of future work, and guide model improvements.)*

**Ref#1**

Although Sect. 3.3.2 explain a bit, I still cannot understand why use air temperature rather than soil temperature. Do you mean soil temperature has very large uncertainty or it requires more coding work to implement?

**Reply:**

*Sect. 3.3.2 provides two explanations. The first reason, denoted (a) on L196, reflects the practical consideration that the soil temperature was not available from the EMEP model's data files. Given the very limited funding provided by CAMS-81 for this work, and the tight deadlines, we built a system around the EMEP model files which were readily available. (The EMEP model data does come from ECMWF, but is the subset needed for our chemical transport model.) The second reason given, denoted (b) on L187, is that soil temperatures are indeed difficult to predict for large grid cells. We provide references to support that assertion.*

*We can note that in the first version (v1.1) of CAMS-GLOB-SOIL we made use of relationships between air and soil temperature taken from YL95 and from the MEGAN code (Guenther et al., 2006). However, closer examination of these relationships showed some problems (as mentioned in Sect. 4.7). This is clearly an issue that needs more attention, and as noted in Sect. 4.7 and the conclusions we will pursue better methods in future.*

*We will add more text on these points in the revised manuscript. (Ref #2 also had some comments on these temperature issues; we will address those in a separate reply.)*

**Ref#1**

The introduction lacks the review of current soil NOx emission algorithm.

**Reply:**

*We do not really understand this comment. Sections 1 (introduction) and 2 (Nitrogen Oxide emissions: background) provide more than 3 pages of background*

*on soil emission algorithms, and refer the reader to much more extensive reviews that exist. We present the basic ideas of YL95, SL11 and Hudman et al (and hence HEMCO and Weng et al), though of course we do not cover all details. Again, we believe ESSD is a journal where the focus should be the description of the dataset being presented, rather than a journal where long reviews of other works should be presented.*

*Nevertheless, if the referee missed something specific we will be happy to add comments.*

**Ref#1**

Line 3: delete degrees

**Reply:**

*Thank you. It will be deleted.*

**Ref#1**

There are many grammar errors, and the writing should be polished.

**Reply:**

*We will endeavour to improve the English.*

**Concluding remarks**

As noted above, we disagree with the criticism of Ref#1 that this manuscript and the dataset lacks sufficient innovation to be acceptable for publication. We hope we have clarified the nature of the manuscript and of the CAMS-GLOB-SOIL v2.2 database. We are happy to add additional comments into the revised manuscript to make our points clearer. We strongly believe that datasets used by Copernicus, EMEP or indeed any Earth System modellers need to be documented, and we believe ESSD is the appropriate journal for such documentation.